# Construction 4.0 Organisational Level Challenges and Solutions

**Orsolya Nagy** [1], **Ilona Papp** [2] **and Roland Zsolt Szabó** [2,*]

[1] Doctoral School of Economics, Business and Informatics, Corvinus University of Budapest, 1093 Budapest, Hungary; orsolya.nagy@uni-corvinus.hu

[2] Department of Marketing and Management, Kautz Gyula Faculty of Business and Economics, Széchenyi István University, 9026 Győr, Hungary; pappi@sze.hu

[*] Correspondence: roland.szabo@sze.hu; Tel.: +36-30-4453744

**Abstract:** The construction industry (CI) is ancient and has evolved along with humanity, yet it has become increasingly inefficient due to fragmentation, the use of traditional solutions and the lack of innovative technologies and methodologies which are no longer sustainable. The Fourth Industrial Revolution has started to transform this industry, and Construction 4.0 (C4) can advocate this change to become a more efficient cyber-physical ecosystem. However, technology alone will not solve all challenges. While research on C4 focuses mainly on technology, management also plays a key role. We asked experienced company executives for their opinions on the digital transformation in the CI. Research proves that it is not just a technology but primarily a management and strategic challenge.

**Keywords:** construction 4.0; digital transformation; organisation; industry 4.0; management; ecosystem; BIM

## 1. Introduction

The Fourth Industrial Revolution has reached the construction industry (CI) [1] and construction organisations [2]. CI has a long history, almost as long as that of humans. The digital transformation hit this highly fragmented [3] ancient industry [4] and demands changes through the CI supply chain [5] and value chain [6]. The new technologies, methodologies [7] and digital solutions [8] bring significant challenges and solutions for every stakeholder. While building information modelling (BIM) is the catalyst of this revolution, it is not the final solution. While organisations, projects [9], management [1] and governments face several challenges, new stakeholders [10] have realised the opportunity in the CI. The CI can delay the change but cannot prevent it.

Construction 4.0 (C4) is an increasingly frequently mentioned and widespread term in social media; however, there is limited research on the exact definition. C4 is mentioned as applying Industry 4.0 (I4) principles/technologies in the CI [4,7,11,12]. The peculiarity of C4 compared to I4 is that it has significantly more uniqueness in it. C4 is a change of approach based on significantly closer collaboration across the entire supply and value chain, supported by industrial processes, methodologies [7,12–17] and innovative tools [4,7,12–15] through a digital platform(s). Digitization can also significantly reduce lead times, costs, environmental impact and carbon emission, but its completion still has significant demand and supply constraints. Nevertheless, previous studies have concluded that the introduction of Construction 4.0 technologies is a major challenge in this generally slow industry, with only 6% of construction companies in 2019 taking advantage of it [18].

The COVID-19 pandemic has pointed out that the workforce in CI is in a particularly vulnerable position, and the effective application of technology [19] and the creation of a virtual work environment [20] providing possible solutions to the current situation. The

overall industry transformation will result in significant efficiencies in addition to the current pandemic, which I4 has already demonstrated in many cases. The purpose of this article is to understand the challenges and solutions of C4. There are such studies in Industry 4.0, but in C4, these have not been explored. Therefore, it is important to understand these factors and what the dominant experts in the CI think about this to achieve a more sustainable industry.

This paper shows the C4 human resources, management, organisation, financial resources, market conditions, and customer satisfaction challenges and solutions. The research is based on 29 semi-structured interviews with experts. Grounded theory approach was used to identify these challenges and solutions. The results show that the biggest obstacle to spread the C4 is the lack of customer demand in significant efficiency gains, especially in the construction phase. A more efficient construction would have a lower environmental impact that has important consequences for construction customers.

The paper is structured as follows. After the introduction, the second part introduces the theoretical background, which introduces the most important literature on C4 and its organisational level challenges and solutions. The third part provides details about the grounded theory approach, the expert selection and classification and the interviews and coding. The qualitative research results are introduced in the fourth part. Finally, the last part includes the discussion of the results and further research directions.

## 2. Theoretical Background

### 2.1. Industry 4.0 (I4) Principles

The technological development brought about by the Fourth Industrial Revolution has had a major impact on both society and the economy, thus significantly impacting corporate competitiveness and social welfare. This phenomenon was first outlined in 2011 [21] and it refers to the integration of information and communication technologies in an industrial environment [22]. In the I4 era (1) digitisation, optimisation and personalisation of production; (2) automation and adaptation; (3) human–machine collaboration; (4) value-added services and warehousing; and (5) automatic data exchange and communication were defined as the five key elements [23]. Zezulka et al. [17] added three more elements to I4: (6) digitisation and network integration, (7) new market models and (8) digitisation of products and services. Integration of technologies can be implemented from different perspectives, such as horizontal (network between organisations), vertical (interrelating technologies within organisations) and end-to-end (interrelating technologies through entire processes) [24,25]. Consequently, digitalisation affects the entire value chain from business models to management systems [26]. In addition, I4 is also expected to lead to more sustainable production, with reduced material consumption and reduced waste [27–29], which is especially important for the CI. The main driving factors and barriers of I4 have been identified as human resources, organisation, management, market condition and competition, financial resources and profitability, productivity and efficiency, customer satisfaction and technological and process integration [29,30]. Expectations for Industry 4.0 include increased productivity, better use of resources and better product quality. In addition to the usual benefits, it also allows flexibility in production, i.e., series production can be economical, even when the size of the series is a single piece [4].

### 2.2. The Concept of C4

I4 can radically improve the CI. The Boston Consulting Group hypothesized in 2016 that the CI will be "soon" characterized by connected systems of sensors, intelligent machines, and new software applications—all integrated on a central platform of building information modelling [31]. Unlike other industries, i.e., manufacturing, construction has been slow to adopt these new technologies [32]. Moreover, many authors conclude that

construction is still rather low-tech, using heavily craft-based methods, hence the implementation of C4 is falling behind expectations and faces great challenges [18,33].

One possible explanation for the slow implementation is that the application of robotics and automation is limited in the CI, because the complexity of the tasks to be performed is much higher than in other industries and the need for dynamic change and adaptation is common in on-site construction projects [7,33]. The CI is often characterised by high diversity of agents who are reluctant to change [11]. Moreover, other factors such as extreme fragmentation and lack of collaboration also limit the implementation of innovative construction technologies [11].

Nevertheless, in this era, the new phenomenon called C4 has appeared and is widely referred in the media, used on social media. The amount of research in C4 is growing exponentially; however, there is still limited research available on the topic. Based on a Scopus keyword search, 9 articles were published between 2015 and 2019, 15 in 2020 and 8 in 2021. In the keyword search, we considered articles whose keywords included "construction 4.0" and then limited the search to "industry 4.0", "construction industry", and the "industrial revolution". In presenting the present theoretical background, we have thoroughly examined the articles explored in the Scopus database and their references.

Consequently, the aim of the article is to foster the implementation of I4 in the construction sector by highlighting barriers and their potential solutions. Although the digitalisation of the CI may bring further solutions, the present research focuses specifically on the phenomena of construction 4.0 and its challenges and solutions.

Despite the limited research on the topic of C4, it is mentioned as the application of I4 principles/technologies in the CI [4,7,11,12]:

1.   C4 as the application of I4 in the CI;
2.   Application of industrial processes and new methods;
3.   Construction-specific innovative tools: devices, information technology, materials;
4.   Industry-wide collaboration between construction professionals, start-ups and digital firms.

Industrialised processes and new methodologies can bring new effective ways to the built environment [34] and C4 is definitely reshaping the CI, making it more attractive [1,11,35]. Additive manufacturing (3D printing), modularisation [14] and off-site construction [7] are new elements of the C4 era compared to I4. BIM methodology is the essence of the C4 environment [7,12–16]. A new method called digital twin construction is emerging by improving BIM combined with lean construction thinking, artificial intelligence, AI and data-based construction management [16]. In addition, a new process, a product life-cycle management that manages the product from design to retirement, was added as a new element of C4 [14].

Construction-specific innovative tools: devices, information technology and materials are additional elements of C4. Robotics [4,7,12–15] and RFID [7,12,14] are mentioned as additional elements of the evolving industry, however, augmented reality, virtual reality [7,12–15] and IoT [12,14,15] were also mentioned as the key devices of C4 despite those also being I4 devices.

Other information technologies are mentioned by previous studies, the following in empirical studies, three of which are worth mentioning. First, mobile computing was mentioned as a new element in construction [14], while artificial intelligence and machine learning [4,7,13,15,16] and big data analysis [7,14,15] both appear in industrial and construction environments. In addition, new materials concerning industrialisation are part of this evolution [15].

New stakeholders in the C4 environment have not been studied. However, Danel et al. [36] introduced C4 as a collaboration based on I4 principles between construction professionals, start-ups and digital firms, which calls for scholarly attention. Former researchers, however, concluded that I4 imposes a new socio-technological challenge that economic actors cannot solve on their own or through traditional inter-organizational

cooperation and learning [37]. Technological disruptions demand a paradigm shift in solving problems and the consequent learning [38–41]. Nevertheless, it is important to highlight that previous studies have focused mainly on the manufacturing industry in order to gain insights about organisational challenges (i.e., the works of Cimini et al. [42] and Veile et al. [43] and Müller et al. [44]).

### 2.3. Organisational Level Challenges behind C4

Cimini et al. [42] argued that understanding and modelling the role of humans is crucial to develop efficient manufacturing systems of the future. In line with their argument, the successful implementation of C4 calls for moving beyond technology in the investigation of barriers to change [38]. Such a barrier is the limited availability of skilled labour, especially in developing countries [45]. While the CI has already struggled with the lack of human resources and labour [15], the effect of the COVID-19 pandemic has brought further mental health and burnout challenges to the industry [46]. New skills are the major challenges to the implementation of C4 [1,7,10,47,48]. Social and mindset change is crucial to prepare workers for the future human and robot collaboration [1,4] that brings new roles and tasks for daily work [7,49]. Furthermore, implementing more and more robots in the industry brings ethical questions and can increase the unemployment rate [15].

Organisations face new challenges with new workflows [1,7] and organisational change [1,50] that technology demands but old business models [10] and hierarchical organisations [51] hindering the transformation. The low innovation culture [4] and the low speed of technology implementation [1] can be accelerated through well-defined digital partnering agreement [2]. SMEs have a small value in applying I4 principles [50].

Management face knowledge and decision problems. The growing number of innovations demand extra technological knowledge [4] and external support [47]. In the absence of data on technology investment, decision making becomes increasingly difficult [52]. On the other hand, the management decision is made even more difficult by the employees [12] who do not value technological innovation.

The high initial **cost** is hindering technological adaptation [1,49,53–57]; meanwhile, technologies that provide expensive training [50] in most cases have a lack of cost–benefit study [49].

Table 1 summarises the main challenges for C4 identified in the literature. The table shows that in the C4 environment, Maskuriy et al. [1] and Muñoz-La Rivera et al. [7] identified most of the challenges. Five articles identified that new skills for the workforce is a challenge. Furthermore, three articles highlighted the need for change in organisational and work processes. The initial high costs were also mentioned in three studies.

**Table 1.** Challenges behind Construction 4.0 (C4) are identified from in the literature.

| Challenge | Sources |
|---|---|
| New skills for human labour | [1,7,10,47,48] |
| Organisational and workflow changes | [1,7,50] |
| Management knowledge in technologies | [12] |
| The high initial cost of new technologies | [1,49,53] |

### 2.4. Organisational Level Solutions behind C4

There is a great deal of debate today about the job-creating or destructive effects of I4 [54]. There is no doubt that the I4 paradigm influences and shapes the professional skills and competencies required in the future [55,56], therefore, its effective implementation requires special attention to the organizational structure [42]. C4 is reshaping the CI, making it more attractive [1,11,35]. The recent pandemic proved the value of human resource efforts, especially in terms of well-being and occupational health in the sector [46]. Technology demands new skills but creates new roles [10] to support knowledge transfer to

robots [11]. Social network analysis can help increase workers' skills and transfer knowledge inside the organisation [51].

Contrary to Newman et al. [50], Lekan et al. [13] discuss SMEs as the beneficiaries of the technologies. Furthermore, digital partnering is a new solution for businesses to successfully implement new technologies [2] and group construction in a new collaborative resource-sharing way [10] Organisations can solve the innovation challenges by involving research and development [1,10,49] by applying cross-functional networks inside especially hierarchical organisations [51] and by implementing the innovations to every level of the organisational strategy [57]. Furthermore, sustainable change can drive further the C4. Although C4 is not mentioned, Henderson et al. [58] found that collaborative organisations in megaprojects can drive this sustainable change by applying learning logic. The method supports strategic and action steps to focus on inclusion and experimentation.

Managers can apply new methodologies. Situational leadership can solve innovation challenges [59]. Scenario planning is a new tool for managers and researchers [10].

Technologies can increase competitiveness in the global market by allowing local companies to step out internationally [2] or increase the local competitive advantages by a new business model such as digital partnering [52].

From a financial point of view, Alaloul et al. [35] suggested activity-based *costing* to use as a tool to "(1) identify the inspection activities; (2) compute the cost driver rate; and (3) to enable the performance of the scenario analysis by manipulating the volume of cost drivers under different scenarios". Sacks et al. [16] emphasised that technologies can reduce the cost of workers and materials. Furthermore, the platform business model can solve complex management problems and solve supply chain, material and equipment issues, leading to cost saving [6].

Table 2 summarises the solutions behind C4 identified from the literature. Most articles highlighted the importance of involving research and development and making the industry more attractive. The main studies on this phenomenon were carried out by Lavikka et al. [10] and Garcia de Soto et al. [11].

**Table 2.** Solutions behind C4 identified from the literature.

| Solution | Sources |
|---|---|
| Make CI more attractive | [1,11,35] |
| New roles and skills | [10,11] |
| Social network analysis | [51] |
| Digital partnering and group construction | [2,10] |
| Involve research and development and innovation | [7,14,15] |
| Situational leadership | [59] |
| Early technology involvement | [11,33] |
| Sustainable solutions | [10,35] |
| Increase competitiveness | [2,52]. |

## 3. Research Methods

This study aims to give a full perspective of the digital transformation in the CI and explores the C4 challenges and solutions. The grounded theory approach was used to identify the organisational challenges and solutions of C4 based on 29 expert interviews.

Grounded theory is a systematic method to explain a particular phenomenon with the increasing information obtained during the curse of research. The method was developed by Glaser and Strauss [60]. The systematic discovery of the theory includes data collection and analysis, process analysis, constant comparison between the data and the emerging theory, the theoretical sampling to confirm the originally formulated categories with the new data and the theoretical coding [61]. During the research, constant iteration was used to identify the final theory.

The semi-structured interviews were conducted between November 2020 and December 2020. The interviews were recorded with Microsoft Teams, and we used Microsoft Office 365 Word for automatic transcription. After the automatic transcription, the correction was necessary to use the transcribed file for further analysis.

Thanks to the development of technologies, new ways exist of getting each interview to experts. In this way, experts were selected based on their experience from their LinkedIn profiles. The following four keyword combinations were used during the LinkedIn search: construction innovation consultant, construction 4.0 consultant, digitalisation construction expert, construction technology. In addition, experts were analysed also based on their publicly available bibliographic information. Snowball technique sampling was applied after the first five interviews to find further experts on the field. The method helps to involve further actors of interest in the research through the reference of people with the appropriate expertise [62]. We aimed to select experts across the world that varied for the following aspects: Experts from different company sizes (multinational enterprises, large domestic companies and small and medium enterprises) from business field, government or education who have a high impact on the CI digital transformation with their action, with experience from the current highest technological advancements. We also examined the current dominant industry for each interviewer that refers to the financial income of the current workplace. Six dominant industries were selected: CI, electronic automotive industry, IT, manufacturing, government and real estate. Anonymity and confidentiality were assured for the interviewees. We highly recommend using this method in further research.

Seven stakeholders were identified in C4, consulting companies supporting the CI with technological or educational knowledge to drive the industry digital transformation further. General contractors can directly experience the challenges and solutions of digital transformation. Research and development are working to speed up the digital transformation with applied research or state-of-the-art business ideas. Investors or real estate investors aim to develop the best building as a product. Authority is a key stakeholder to speed up domestic industry transformation with local regulations. Students can bring a generational change to the industry. Associations can support knowledge sharing across countries and industries. We identified the suppliers and the technology providers who transform the industry with innovative material or digital technology. Another new stakeholder, the technology investor, supports and invests in technology providers such as industry-specific start-up hubs. Table A1 in the Appendix A shows the details of companies and the interviewees involved in the research.

An interview guideline was formulated to help us navigate the research. During the development of the qualitative research, Kvale's [63] recommendations were applied. The interviews consisted of four main parts. In the first part, an interviewer started by briefing the topic. The interviewee's experience was discussed in the second part: main expert fields and current and past activities in the CI. In the third part, three main topics were discussed. First, the meaning of construction 4.0 and how they see the digital transformation in the CI and its main challenges and solutions. Second, what their vision is for the future of CI. Third, what is their knowledge and experience in different innovations? Finally, the interview finished with a short debriefing.

QSR NVivo software supported coding and in-depth text analysis to apply grounded theory during the data analysis. Coding was divided into three phases: open, axial and selective coding. During the open coding phase, we used automatic coding for each transcription to understand the key terms. We found some connections between codes in the axial coding phase, and we organised similar concepts into groups. We merged our original coding concept based on initial research with the open codes to create a coding tree in this phase. We used the new coding tree to code each paragraph again. A mind map was created at the end of this phase to understand the possible research directions. Finally, sub-categories were identified in the selective coding phase.

## 4. Results

### 4.1. Challenges and Solutions of C4

The results of qualitative research have shown that there are several challenges at each organisational level in the C4 environment that require different approaches. Based on the suggestions of the 29 experts, we explored several possible solutions. The presentation of the results was divided into six main parts. Following the summary table, we present in detail the challenges and solutions offered by C4 in the following order: human resources and society, organisation, management reality and mindset, market condition and competition, financial resource and profitability and finally, customer satisfaction. The challenges and solutions found in each organisational category are illustrated in Table 3.

**Table 3.** Challenges and solutions behind C4

| Challenge | Category | Solution |
|---|---|---|
| Attractive industry | | Situational awareness |
| Education | | Young generation |
| Skill and labour shortage | Human resource and society | Reform the industry brand |
| Human feeling | | Low skilled labour |
| Generational issues | | Knowledge transfer |
| Robot and human collaboration | | More social industry |
| Information and data | | Faster adaption of SMEs |
| Innovative mindset | | Data handling in the contract |
| Historical assumptions | Organisational factors | Data collection |
| Procurement and bidding | | Data sharing |
| Collaboration | | R&D investment |
| Technology and process | | |
| Technology pressure | | |
| To see the transformative industry | | |
| Organisational politics | | |
| Understanding the value of innovation | Management reality | |
| Believing technology without process | | Think collaboration as a profit |
| Mindset change | Management expectation | Change decision-making process |
| Lack of training | | Strategic thinking |
| Technology competition | | New business model |
| Cheaper technologies | Market conditions and competition | |
| Governmental level competition | | |
| Single digital market | | |
| SMEs market position | | |
| Low margin | | Start-up solutions |
| Lack of process innovation leads to waste of money | | Balancing the budget with technology |
| Cost of labour | Financial resource and profitability | New business model |
| Cost of robots | | The monetary value of handover |
| Cost of subscription | | Using available technology |
| | | Cost of connectivity |
| | | Material saving |
| Time pressure | | Building as a product |
| Improving customer satisfaction | | Building as a service |
| Demand for quality | Customer satisfaction | Space as a service |
| Lack of customer demand | | Sustainable solutions |
| Changing the bidding process | | Collaboration platform |

### 4.1.1. Human Resource and Society

The first factor is the human resources and society group. Human resources face extreme challenges in the CI with more and more labour and a massive skill shortage. Furthermore, the sector has become less popular and less attractive to younger generations. Lack of career paths, gender problems and generational problems inhibit the industry's attractiveness. These factors can lead an entire generation to never even entering the industry. Emerging technologies and methodologies force every stakeholder to continuously educate their workers and gain new computer skills, which is exceptionally challenging for the older generation. Human feelings and behaviour tend to inhibit technological and methodological changes. There are a lot of inherent behaviours, all of which will stop organisations from transforming. Workers need to unlearn their inherent behaviours. Another factor is the appearance of robotics on the construction site. A highly trusted environment network and psychological safety are necessary to develop between humans and machines. Robots will open new job opportunities but will also increase the unemployment rate in the sector.

> *"The construction industry is seen as an industry that doesn't really have career paths. That's partly because there aren't always many kinds of joins to upskill people more, enable people to move within the industry to different disciplines." (Interviewee 15)*

> *"Several Member States there are the facing a lacking workforce or in if they have an ageing workforce and we see that the constructions it is not attractive for younger generations." (Interviewee 8)*

> *"One interesting point is, especially in Africa, that is lack of trained excellence, to use some of those technologies." (Interviewee 14)*

> *"Also, for the consultants, things have become more complicated, and they've got a problem with the skill set. That thing risk now in producing these construction details, they end up getting things wrong, and people see them, but at the same time there's lots of good money in that." (Interviewee 22)*

> *"90% of construction businesses at least are already paying for Word, Excel, and PowerPoint, and then you just teach them how to work on the same document at the same time, which is BIM." (Interviewee 26)*

> *"Construction, as an industry, is still learning that the industry needs to become more social. Construction has growing up to do in terms of the way that they treat their people, and I think that we're going to see a shortage of tech strong workforce because they're being pulled out by the technology companies." (Interviewee 20)*

Experts believe that technological advancement can make the industry more attractive. Organisations tend to forget that not only the young generation is interested in new technologies. Engineers get very excited by technologies. People in the sector are also very visual. For instance, AR/MR or VR can support visualisation in the office environment and construction sites. New tools and real-time information can support situational awareness and increase workers' responsibility. The opportunity to implement new solutions should be given to talented and energetic people. A more social environment with a broader career horizon can increase workers' engagement. New technologies attract the new generation of workforce. Knowledge transfer from other industries from automotive and manufacturing can speed up the change. Broader experience can bring new concepts to organisations. Human resources need to focus on recruiting IT staff such as coders and developers to work with engineers who add high value to the business. Individual self-training can support experts to find new job opportunities. Less skilled labour can be expected in the controlled prefabrication environment because fewer and fewer highly skilled people are on site.

> *"Not just in the younger people are more attracted by anything that is digital electronics, and if the sector is completely lacking in that field is not just there, not sexy for them so that they go to other sectors." (Interviewee 8)*

### 4.1.2. Organisational Factors

The second group of factors is the organisational factors. Results showed that information and data collection and sharing bring further problems to the surface. To find relevant information in the growing number of data demands expertise. Furthermore, trust is a crucial element in information sharing. Lack of trust between and inside the organisations hinder their ability to collaborate. Disciplines have different preferences, and the growing number of technologies and platforms does not support collaboration either. It is still an automatic individual standard on how to collaborate. Redesigned and continuously updated contract templates can enforce collaboration. Suppliers are struggling to sell the emerging number of innovations while procurement and bidding departments still have not incorporated the innovation's cost saving into their processes. Historical assumptions and innovative mindset hinder the organisation from growing and from implementing technologies. MNEs can be inhibited from technology innovation by an enterprise agreement, internal political situations and a long decision-making process. Organisations should prepare to understand how to choose and connect digital interfaces. Technological advancement is still critical without changing the current workflow.

*"You know your competitive advantage should not be the data about the buildings you're constructing." (Interviewee 9)*

*"I think the implementation is the main issue. The process is not easy as well, but implementation is the toughest thing." (Interviewee 5)*

Data collection can provide useful information to organisations for the maintenance of tools and equipment. Proper data handling and data collection can support better contracts. Trusted, authenticated, and stacked data from third-party agencies can increase trust between organisations. Data handling can be a part of the contract to increase collaboration and transparency between contractual parties. Investing in research and development can speed up organisational transformations and technology implementation. Proper evidence for the return on investment in innovation is necessary for decision makers. The changing industry brings solutions for SMEs such as faster innovation implementation due to size and flexibility

### 4.1.3. Management Reality and Expectations

Top management refuses reality, the transformation of the industry. They do not understand the technologies and methodologies that put high pressure on them. Lack of knowledge and training leads to an unclear understanding of the benefit of the technologies and the financial and social value of the innovation. They believe their business is efficient and constantly evolving as old working methods are still profitable. They are afraid to take risks posed by innovations, which would reveal organisational policy issues and long-term problems within the organisation. Rejective behaviour often leads to the killing of the enthusiasm of ambitious employees. Due to the lack of management knowledge, the transformation of internal processes after the technology investment is missing or delayed, leading to further problems and tensions within the organisation. Technological change requires extra knowledge, and for that, education is necessary and must be accepted. Innovation changes and decision making depend on top management, but change in management also requires pushing the lower management level.

*"I think the pressure on the management, especially on the top management of large construction companies, will certainly increase over time." (Interviewee 4)*

*"They cannot say because they do not understand what is possible, so it is hard for them to see the big picture." (Interviewee 11)*

*"I think that then they will invest in some of their work for security or have to be trained to do this. I think this is very important. If they do not, if they are not convinced that they will have a benefit from new technology, they will just continue business as usual." (Interviewee 8)*

Infusing research and development can support mindset change and help with the current status and problems in the future. By leveraging new business models, general contractors can be a part of what they created. Technologies can support new business models to handle building as a product, building as a service or space as a service. By thinking of collaboration as a profit, decision making and strategic thinking can be improved. Furthermore, technological implementation increases the quality and delivery of the service and change the organisation market position.

*"So we are moving from this idea of isolated or standalone products. We are moving towards products and services." (Interviewee 7)*

### 4.1.4. Market Conditions and Competitions

The market is evolving in the direction of a single digital market. A growing number of technologies are replacing traditional methods. Laser scanning is becoming a fundamental tool in the market; well-developed robots are appearing. Cheap sensors are connecting interfaces. Technology providers are starting to share the market, and stronger competition will speed up the improvements. The implementation is increasing in the build and operation phases. SMEs can change their market position by fast technological adaption. The market will expand, and it will be easier for businesses to exchange information. Data maintain a competitive edge; however, keeping closed information can lead the company to market fall. Governments are developing country-level strategies in order to keep the country internationally competitive and to centralise digital information.

*"The traditional measuring methods are slowly, slowly becoming obsolete because there's simply a technology becoming cheaper and cheaper." (Interviewee 28)*

### 4.1.5. Financial Resource and Profitability

Medium-sized companies right down to micros in Tier 3 have a lower margin, which is not particularly conducive to investing in digital technology. Calculating the return on investment to find the balance between cost and time is challenging, especially for SMEs. Some technologies have an expensive subscription price that leads to missing stakeholders in the BIM life cycle. In some countries, a cheap labour force is blocking technology adaptation, while in others, unsocial environments can increase the cost of labour, leading to an increase in the construction cost. The price of robots needs to drop significantly to be able to spread within the industry. Using innovative technology in the old way is a waste of money and can generate even further expenditures.

*"Subscription, and it's ridiculously expensive and less prohibitive." (Interviewee 10)*

*"Labour rates will rise because we have to pay people more in order to get them interested. So then construction costs of construction will go up because there's a direct relationship with the labour cost." (Interviewee 20)*

C4 offers many cost-saving solutions. By organising the projects smartly it is possible to make savings 50% of the time. A small marginal project that does not have the resources or budget to make mistakes can take the digital leap to take the full advantage of technology by balancing the budget and schedule with safety and quality. SMEs and investors can save costs for the whole project budget by early-phase initial investment. The cost of connectivity is almost free; the internet is now an essential utility. SMEs are already subscribing to cloud-based office applications. Using business information management processes, the cloud-based solutions provided by those applications could effectively help with collaborative work and process optimisation without extra financial investment. The exploding start-up market shows the value of clash detection.

Furthermore, generative design applications can save a significant amount of raw material, leading to cost savings. The value of structured data is increasing. The handover documentation represents an increasing monetary value as it reduces the operating expenses, which can be 78–90% of the total investment. Building design and construction

are under 10% of the overall building cost and 90% of the operational cost. New business models can generate higher profit for businesses that are currently focusing only on the construction phase.

> *"Without any technology, just by organising the work smartly, you can save 50% of all the time and using technology will boost it even further." (Interviewee 16)*

> *"Handover is really important. So it's the most in terms of monetary value. It's one of the most important to get right because 78% of your OpEx costs around." (Interviewee 23)*

> *"Why not be part of the ongoing money? Because if you look at the pie of the life cycle of a building design and construction is under 10% of its overall cost. The real cost is in operations. That means 90% of the value of that property." (Interviewee 3)*

### 4.1.6. Customer Satisfaction

The sixth factor examined is customer satisfaction. Government is one of the largest customers in the built environment, but private investors also play a key role. Customers do not require high-quality materials, methods and technologies at every stage of the construction value chain. An additional challenge is that potentially changing customer requirements are further hampered by old tendering processes, which further reduces the possibility of early-phase innovation and sustainable solution involvement. Rising material prices and time pressures further increase costs, which leads to the lack of customer demand.

> *"Once you're able to separate them and holistically, look at the building and then you're able to optimize it and change things and understand exactly the implications. So it's it's about looking at the product and building a better product." (Interviewee 18)*

> *"The group of people that will profit most from all those developments are the owners. Once the governments realise that all those things would benefit them the most, they will start to increase, fostering and demanding building information modelling and other technologies and practices." (Interviewee 4)*

New methods and business models support the growth of customer satisfaction. A centralised platform can provide a solution to transform traditional processes that integrates the entire building life cycle and the circular economy. Building as a product involves a holistic approach where the building is examined and constructed broken down into its elements. Building as a service is not just a point of view; it also involves a whole new business model. In this case, each function of the building is considered the highest level of service sold to the customer. This kind of attitude extends to the operation phase of the building, where the service provider continuously uses the highest level of materials and technologies in the given building. Space as a service is not a completely new model, which further improves the concept of mobile homes and provides a given area for customers with adequate infrastructure. In addition, sustainable solutions can further increase customer satisfaction. Continuous monitoring of the value of building materials used in the building under construction appears as an additional source of revenue for the customer.

## 5. Discussion and Future Research Directions

The rapid development of innovations will lead to a complete transformation of the CI and bring many challenges and solutions. Researchers face an entirely new area where, in contrast to the industrial environment, the constant change of locations and actors requires a unique solution. Although an increasing amount of research has been conducted on the implications of deals with new technology. However, limited research has focused on its deals with implications for the organisational issues. This research examines the following organisational level factors: human resources, organisation, management reality and mindset, market conditions and competition, financial resources and profitability,

and customer satisfaction. Our goal was to formulate the challenges and possible solutions at these levels based on the opinion of market stakeholders and the literature review. The research supplemented the literature background at all examined levels.

Theoretical background and our research confirmed that the decreasing number of available workforce is not the only challenge for human resources. Innovation demands social and mindset change [1,4] to adapt new skills [1,7,10,47,48], especially for human-robot collaboration that brings further ethical questions [15]. Furthermore, we found that training and education at all organisational level is necessary to transform the labour force in preparation for the implementation of C4 and the CI successfully. In addition, social network analysis can help existing employees find the right place for them in the company [51] The results of the qualitative research indicate that the repositioning of the current brand of the entire CI could positively influence the continuous supply of labour.

Technologies put solutions in the hands of fast-growing businesses that can save multiple human resources. Results also support current research and theoretical background support that innovations that create new positions make the CI more attractive [1,11,35]. In addition, social network analysis can help existing employees find the right place for them in the company [31]. The results of the qualitative research indicate that the repositioning of the current brand of the entire CI could positively influence the continuous supply of labour. The jobs will be less routine and will demand from the labour force computing and digital competence as well as teamwork, decision making and critical thinking.

The digital transformation requires organisational changes [1,50], however, the innovation culture in the CI is relatively low [4]. Research has confirmed that historical assumptions and the lack of innovative thinking inhibit transformation. The hierarchical organisational structure [51] and old business models [10] provide an additional barrier to innovation. Research has also indicated that change is hampered by existing workflows [1,7]. The research results also pointed out that, in parallel with workflows, cooperation within the organisation is a problem, which is further worsened by the lack of trust and lack of shared individual standards and the lack of trust. The rapid implementation of technologies is a challenge for organisations [1,64], while SMEs have a small value in applying I4 principles [50]. The research results revealed that the growing number of technologies and platforms, lack of expertise, synchronisation of technology collaboration, long decision-making process, and organisational policy further slow down this process. Regarding internal processes, the value of innovation is not taken into account in procurement processes. The constant lack of updating of contract templates further weakens co-operation in an innovative environment. In addition, MNEs can be inhibited from technology innovation by an enterprise agreement.

The current research confirms that organisational transformation can be facilitated by research and development [1,10,49]. An organisational strategy considers innovation at all levels [59] and cross-functional network support technology implementation [51]. Collaborative organisations can leverage learning logic in megaprojects [58]. The current research has revealed that the way data are handled has a significant impact on the organisation. Collecting the right data facilitates collaboration within the organisation, operation, transparency and better quality of contracts. Confidence between organisations can be further strengthened by data purchased from third parties. Furthermore, decision-making processes can be facilitated and accelerated by appropriate case studies on return on investment. Although I4 has less value for SMEs [50], the implementation of innovations can be much faster and more efficient.

Due to insufficient technological knowledge [4], it is challenging for management to decide on innovations [52], which both theory and research have supported. In addition, research has shown that one reason for this is the need for a mindset change. Management does not recognise that the CI is transforming, and they accept that the business is profitable without sustainable changes. In addition to organisational policy, new technologies

put enormous pressure on the other management layer, but some do not understand the need to change processes during implementation.

The theory has shown that the application of situational leadership can solve problems regarding innovations [59]. In addition, this research confirms that changing decision-making processes and strategic thinking is also an efficient tool in the rapidly changing environment. Finally, research interviews have confirmed that collaboration is a solution for management if they think of it as a profit.

There is a lack of cost analysis studies between the high initial investment costs of innovations [1,49,53] and training [50] that both the theory and this research have found. In the absence of a cost–benefit study, this significantly segmented industry and its leaders see a much greater risk in using an unknown technology. Furthermore, technology transition brings ongoing additional costs such as organisational transformation, process transformation and education. Furthermore, our research has shown that the cost of labour is increasing, and the cost of subscribing to online services is often much higher than what is affordable for SMEs in the market. In addition, the implementation of innovations without process innovation explicitly results in additional costs for organisations.

Based on the theoretical background, activity-based costing [35] and this research have also found that innovative technologies and new business models can effectively reduce labour and material costs [16]. This research has also found that companies make little use of the technologies currently available to them, and for construction projects, it would be possible to balance the budget with the technologies that start-up solutions offer. In addition, the monetary value of handover represents an entirely new value for the CI.

Construction 4.0 brings new market opportunities. Competitiveness is supported by new business models [32] and innovations enable local companies to rise internationally [3]. Research has shown that greater competition further accelerates the development of innovations, and a growing market further facilitates the exchange of information. Furthermore, the exploding industrial revolution opens new avenues for resilient SMEs as they can adapt more quickly and flexibly to innovation solutions; this will not only be an improvement for SMEs but may even lead to a complete reorganisation of the market. In addition, centralised digital state-level strategies will further help companies' international market position.

In addition, the sixth organisational factor examined is customer satisfaction. Organisational level challenges revealed that, as opposed to I4 [30], the biggest obstacle for C4 is the lack of customer demand leading to value-chain inefficiency and serious environmental footprint. One of the largest customers in the construction environment is the government, which participates in the entire building life cycle and can bring C4, a sustainable change to the built environment. A centralised platform can support the C4 transformation that integrates the entire building life cycle and the circular economy.

## 6. Conclusions and Future Research Directions

Innovator organisations that take technological risks to make their business more efficient can become an increasingly attractive place to work for new generations. However, given the size of businesses, the task is becoming increasingly complex for larger businesses, including business model change, organisational transformation, process change and technology collaboration.

The managerial implications of the study are twofold. First, it is recommended to maintain a parallel focus on both socio- and technical factors of the organisation. Second, the management can minimize ambiguity of the digital transformation by closely studying other industries. The manufacturing industry has already documented transformation models which could serve as a guideline.

The lack of innovation knowledge creates uncertainty for management; therefore, we recommended involving consultants and universities with relevant experience to reduce the risks inherent in such a big-scale transformation based on the research. The

hierarchical organisational structure and centralised decision-making processes hinder innovation. Thus, rethinking the transformation of the organisational hierarchy and introducing decentralised decision-making processes can support technological changes. A review of current processes can help identify points where innovations need to be implemented. Technology management is not necessarily the responsibility of the IT department. Creating an innovation manager or an entire innovation department is worthwhile based on the organisational sizes. It is worth noting that experience gained from the automotive and manufacturing industries can bring particularly high value to the company for these positions.

Furthermore, the benchmark plays a significant role in industry transformation. Actors outside the CI can act as catalysts in businesses, and other industries' experience can make processes more efficient. The innovation department within the organisation can monitor the innovations and coordinate their implementation and education at all levels. In addition, it can continuously monitor the implemented innovation. It is also worthwhile to involve existing employees in selecting innovations, thereby reducing the tension within organisations arising from technologies. After each innovation is introduced, reviewing and monitoring existing processes can help to improve internal processes. Innovation-based cost analysis can also significantly reduce innovation risks and support the decision-making process. Organisations can further reduce the risk if the companies test the innovations in a consortium or together with research universities and consultants in the framework of pilot projects and then implement them for further projects.

The exploding industrial revolution opens new avenues for resilient small businesses as they can adapt more quickly and flexibly to innovation solutions; this will not only be an improvement for small businesses but it may even lead to a complete reorganisation of the market. Technologies put solutions in the hands of fast-growing businesses that can save multiple human resources. An SME layer that cannot evolve with technology needs external support. Research transfer with non-profits' support and governmental finance can give them solutions.

Finally, we see that technological advances and the emergence of ecosystems are inevitable. Technology connects the industry on a national level, and the industrial level can form an ecosystem. Therefore, the authorities are a key player that needs to be involved in digital transformation processes soon. Furthermore, the authorities are the biggest customer and a stakeholder involved in the whole life cycle, and they could benefit the most from continuous development through integrated platforms and the concept of a circular economy.

Our research indicates several new research directions, which could be studied in follow-up research. For example, the social life cycle assessment in the CI could enrich our current understanding of how to progress further. Moreover, the study raised questions at the organisational level. It is also critical for human resources to understand how to make the industry brand more attractive for highly skilled workers. At the management level, methods and tools that can help increase management knowledge and competence should be considered. At the organisational level, it is important to know what kind of organisational transformation can support the C4 and the new business models that speed up this transformation. Profitability could be a significant driver for C4; however, the appropriate cost analysis of innovations requires further research.

Results indicate further in-depth research to reveal the challenges and solutions for C4 at the industrial and government levels. Finally, to reveal the C4 ecosystem, we encourage researchers to explore technologies' challenges and solutions throughout the construction value chain.

## Appendix A

**Table A1.** Details of companies involved in interviews.

| ID | Dominant Industry | Stakeholder in Construction | Country | Organisation Type | Position | Business exp. |
|---|---|---|---|---|---|---|
| 1 | Construction | Consulting/Technology investor/Association | United Kingdom | MNE/SME/NGO | Manager/CEO/Founder | 5+ |
| 2 | Construction | Technology provider | USA | DLC | Director | 10+ |
| 3 | Real estate | Consulting/Technology investor/Investor | USA | DLC | CEO | 25+ |
| 4 | Construction | Technology investor and provider | Germany | DLC | CEO | 15+ |
| 5 | Construction | Technology investor | Israel | DLC | CEO | 25+ |
| 6 | Construction | Research and development | Portugal | EDU | Employee | 5+ |
| 7 | Real estate | Research and development | Finland and India | EDU | Associate Professor | 15+ |
| 8 | Government | Authority | Poland | GOV | Employee | 5+ |
| 9 | Government | Authority | Estonia | GOV | Director | 5+ |
| 10 | Electronic automotive | Investor | United Kingdom | MNE | Manager | 15+ |
| 11 | Construction | Designer | Finland | MNE | Director | 15+ |
| 12 | Construction | Consulting | United Kingdom | MNE | Employee | 5+ |
| 13 | IT | Investor/Technology provider | USA | MNE | Director | 15+ |
| 14 | Construction | Consulting | Nigeria | NGO | Employee | 5+ |
| 15 | Construction | Research and development | United Kingdom | NGO | Manager | 5+ |
| 16 | Construction | Consulting | Finland | SME | CEO | 35+ |
| 17 | Construction | Consulting | United Kingdom | SME | CEO | 10+ |
| 18 | Construction | Technology provider | USA | SME | Manager | 5+ |
| 19 | Construction | Technology provider | USA | SME | Manager | 5+ |
| 20 | Construction | Consulting | USA | SME | CEO | 20+ |
| 21 | Construction | Consulting | USA | SME | CEO | 25+ |
| 22 | Construction | Consulting | United Kingdom | SME | Manager | 20+ |
| 23 | Manufacturing | Technology provider | United Kingdom | SME | Manager | 5+ |
| 24 | Construction | Consulting | Netherland | SME | CEO | 15+ |
| 25 | Construction | Consulting | United Kingdom | SME | CEO | 20+ |
| 26 | Construction | Consulting | United Kingdom | SME | Director | 15+ |
| 27 | Construction | General contractor and education | India | SME/EDU | Manager/lecturer | 5+ |
| 28 | Construction | Consulting | Latvia | SME/EDU | CEO | 10+ |
| 29 | Construction | Consulting | Hungary | SME/EDU | CEO/researcher | 5+ |

Notes: EDU = education, GOV = government, MNE = multinational enterprise, DLC = domestic large company, SME = small and medium enterprise.

**Author Contributions:** Conceptualisation, R.Z.S.; methodology, R.Z.S.; formal analysis, O.N.; investigation, O.N.; writing—original draft preparation, O.N. and R.Z.S.; writing—review and editing, O.N., I.P. and R.Z.S.; visualisation, O.N.; supervision, R.Z.S. All authors have read and agreed to the published version of the manuscript.

**Funding:** This research received no external funding.

**Institutional Review Board Statement:** Not applicable.

**Informed Consent Statement:** Informed consent was obtained from all subjects involved in the study.

**Data Availability Statement:** Not applicable.

**Conflicts of Interest:** The authors declare no conflict of interest.

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
