# Peer review of "Construction 4.0 Organisational Level Challenges and Solutions"

_sustainability, doi:10.3390/su132112321_

Round 1

Reviewer 1 Report

Dear authors, thank you for your submission to Sustainability. Please find my comments below:   Please derive even better the role Industry 4.0 plays for the construction industry and how this industry is different from, e.g., the manufacturing industry. Then, the relevance and research gap becomes clearer.   A conclusion highlighting the main contributions to theory, managerial implications, and research outlook is missing.   Please relate the article to works on organization, people dimensions, etc. already published and discuss the results referring to different industry sectors, such as:   Cimini, C., Boffelli, A., Lagorio, A., Kalchschmidt, M., & Pinto, R. (2020). How do industry 4.0 technologies influence organisational change? An empirical analysis of Italian SMEs. Journal of Manufacturing Technology Management.   Veile, J. W., Kiel, D., Müller, J. M., & Voigt, K. I. (2019). Lessons learned from Industry 4.0 implementation in the German manufacturing industry. Journal of Manufacturing Technology Management.

Author Response

Dear Reviewer,

First of all, thank you for your comments. We improved the article according to your requests.  We added 25 new references to the manuscript. Section 2.1 on Industry 4.0 and other theoretical sections have been expanded with the proposed literature and many more, and the research gaps became clear accordingly.

The article was expanded with a detailed explanation of the managerial implications of the results and the Conclusion part was also added.

Reviewer 2 Report

Dear Authors,

the research and study presented is interesting and deserves attention. The idea of a "Construction 4.0 Organisational Level Challenges and Solutions", it is an important issue especially in this pandemic period. 

However I recommend the following changes to improve the card.
Introduction:
Add more references to highlight the problem more and make it more interesting. What is the research context? What caused the Covid? it is necessary to answer these questions clearly

Method: is the applied method new? Or was it taken from literature? Since we are talking about impacts on the economy and on people, why not also introduce the life cycle cost and the social life cycle assessment?

Conclusion: The conclusions chapter is not present. I have read the results and discussions chapter but it is also necessary to include the conclusions

Author Response

Dear Reviewer,

First of all, thank you for your comments. We improved the article according to your requests.  We added 25 new references to the manuscript.

We expanded the introduction section and explained the effect of COVID-19.

Regarding your questions about the methodology, Grounded theory method was developed by Glaser & Strauss; however, thanks to the development of technologies, new ways are approaching how to get in contact with experts. Falling this path, experts were selected based on their experience from their LinkedIn profiles that is a novel way to connect to experts.

Life-cycle cost and social life cycle assessments are beyond the research scope however, we emphasised the importance that in the construction industry, it could enrich our current understanding of how to progress further with the research.

As you suggested, Conclusion part has been added to the article.

Reviewer 3 Report

First of all, I want to congratulate you on your work on this paper. This very interesting paper presents a fresh and new approach to the Construction 4.0 conception from the managerial point of view. However, to improve the quality of the paper, authors can rethink several aspects: - I am missing the primary purpose of the article in the Introduction part, - The second part, Theoretical Background, has four relatively short subchapters. Maybe, it is worth canceling these subtitles and leaves that text in one. Also, this part ends with a table – please add some text behind it. - Also, I have the feeling that the literature part should be extended – the list of 42 literature sources is relatively short. In this case, authors can leave the subchapters, but they must be extended.  - The research methodology is well described. - Part 4. Results cannot start with a table, a short introduction should follow it - Also, it cannot ends with citations - Maybe it is worth separating a Discussion part and future research direction into the Discussion part and Conclusion part, in which authors will indicate the future research direction, together with limitations of the study.

Author Response

Dear Reviewer,

First of all, thank you for your comments. We improved the article according to your requests.  We added 25 new references to the manuscript, extended the descriptions of the tables and added a separate Conclusion part to the manuscript.

Round 2

Reviewer 2 Report

Dear authors,
the article has been improved following the reviewers' indications. The introduction and the conclusions have been improved. 

the paper in this form can be acceptable for the publication in this journal.